# Inexact trust-region algorithms on Riemannian manifolds

**Hiroyuki Kasai**
The University of Electro-Communications
Japan
kasai@is.uec.ac.jp

**Bamdev Mishra**
Microsoft
India
bamdevm@microsoft.com

## Abstract

We consider an inexact variant of the popular Riemannian trust-region algorithm for structured big-data minimization problems. The proposed algorithm approximates the gradient and the Hessian in addition to the solution of a trust-region sub-problem. Addressing large-scale finite-sum problems, we specifically propose sub-sampled algorithms with a fixed bound on sub-sampled Hessian and gradient sizes, where the gradient and Hessian are computed by a random sampling technique. Numerical evaluations demonstrate that the proposed algorithms outperform state-of-the-art Riemannian deterministic and stochastic gradient algorithms across different applications.

## 1 Introduction

We consider the optimization problem

$$\min_{x \in \mathcal{M}} \ f(x), \tag{1}$$

where $f : \mathcal{M} \to \mathbb{R}$ is a smooth real-valued function on a *Riemannian manifold* $\mathcal{M}$ [1]. The focus on the paper is when $f$ has a *finite-sum* structure, which frequently arises as big-data problems in machine learning applications. Specifically, we consider the form $f(x) \triangleq \frac{1}{n} \sum_{i=1}^{n} f_i(x)$, where $n$ is the total number of samples and $f_i(x)$ is the cost function for the $i$-th ($i \in [n]$) sample.

Riemannian optimization translates the constrained optimization problem (1) into an unconstrained optimization problem over the manifold $\mathcal{M}$. This viewpoint has shown benefits in many applications. The principal component analysis (PCA) and subspace tracking problems are defined on the *Grassmann* manifold [2, 3]. The low-rank matrix completion (MC) and tensor completion problems are examples on the manifold of *fixed-rank* matrices and tensors [4, 5, 6, 7, 8, 9, 10]. The linear regression problem is defined on the manifold of the fixed-rank matrices [11, 12]. The independent component analysis (ICA) problem requires a whitening step that is posed as a joint diagonalization problem on the *Stiefel* manifold [13, 14].

A popular choice for solving (1) is the *Riemannian steepest descent* (RSD) algorithm [1, Sec. 4], which is traced back to [15]. RSD calculates the *Riemannian full gradient* $\mathrm{grad} f(x)$ every iteration, which can be computationally heavy when the data size $n$ is extremely large. As an alternative, the *Riemannian stochastic gradient descent* (RSGD) algorithm becomes a computationally efficient approach [16], which extends the *stochastic gradient descent* (SGD) in the Euclidean space to the general Riemannian manifolds [17, 18, 19]. The benefit of RSGD is that it calculates only *Riemannian stochastic gradient* $\mathrm{grad} f_i(x)$ corresponding to a particular $i$-th sample every iteration. Consequently, the complexity per iteration of RSGD is *independent* of the sample size $n$, which leads to higher scalability for large-scale data. Although the iterates generated by RSGD do not guarantee to decrease the objective value, $-\mathrm{grad} f_i(x)$ is a decent direction in expectation. However, similar to SGD, RSGD suffers from slow convergence due to a *decaying stepsize* sequence. For

this issue, *variance reduction* (VR) methods on Riemannian manifolds, including RSVRG [20, 21] and RSRG [22], have recently been proposed to accelerate the convergence of RSGD, which are generalization of the algorithms in the Euclidean space [23, 24, 25, 26, 27, 28]. The core idea is to reduce the variance of *noisy* stochastic gradients by periodical full gradient estimations, resulting in a linear convergent rate. It should, however, be pointed out that such Riemannian VR methods require *retraction* and *vector transport* operations at *every iteration*. As the computational cost of a retraction and vector transport operation is similar to that of a Riemannian stochastic gradient computation, Riemannian VR methods may have slower wall-clock time performance per iteration than RSGD.

All the above algorithms are *first-order* algorithms, which guarantee convergence to the *first-order optimality condition*, i.e., $\|\mathrm{grad}f(x)\|_x = 0$, using only the gradient information. As a result, their performance in ill-conditioned problems suffers due to poor curvature approximation. *Second-order* algorithms, on the other hand, alleviate the effect of ill-conditioned problems by exploiting curvature information effectively. Therefore, they are expected to converge to a solution that satisfies the *second-order optimality conditions*, i.e., $\|\mathrm{grad}f(x)\|_x = 0$ and $\mathrm{Hess}f(x) \succeq 0$, where $\mathrm{Hess}f(x)$ is the Riemannian Hessian of $f$ at $x$ [29]. The *Riemannian Newton* method is a second-order algorithm, which has a *superlinear local* convergence rate [1, Thm. 6.3.2]. The Riemannian Newton method, however, lacks global convergence and a practical variant of the Riemannian Newton method is computationally expensive to implement. A popular alternative to the Riemannian Newton method is the *Riemannian limited memory BFGS* algorithm (RLBFGS) that requires lower memory. It, however, exhibits only a linear convergence rate and requires many vector transports of curvature information pairs [30, 31, 32]. Finally, the *Riemannian trust-region* algorithm (RTR) comes with a global convergence property [1, Thm 7.4.4] and a superlinear local convergence rate [1, Thm. 7.4.11]. It can alleviate a poor approximation of the local quadratic model (e.g., that the Newton method uses) by adjusting a *trustable* radius every iteration. Considering an $\epsilon$-approximate second-order optimality condition (Def. 2.1), RTR can return an $(\epsilon_g, \epsilon_H)$-optimality point in $\mathcal{O}(\max\{1/\epsilon_H^3, 1/(\epsilon_g^2\epsilon_H)\})$ iterations when the true Hessian is used in the model and a second-order retraction is used [33]. On the stochastic front, the VR methods have been recently extended to take curvature information into account [34]. Although they achieve practical improvements for ill-conditioned problems, their convergence rates are worse than that of RSVRG and RSRG.

A common issue among second-order algorithms is higher computational costs for dealing with exact or approximate Hessian matrices, which is computationally prohibitive in a large-scale setting. To address this issue, *inexact* techniques, including *sub-sampling* techniques, have recently been proposed in the Euclidean space [35, 36, 37, 38, 39]. However, no work has been reported in the Riemannian setting. To this end, we propose an inexact Riemannian trust-region algorithm, inexact RTR, for (1). Additionally, we propose a sub-sampled trust-region algorithm, Sub-RTR, as a practical but efficient variant of inexact RTR for finite-sum problems. The theoretical convergence proof heavily relies on that of the original works in the Euclidean space [37, 38, 39] and the RTR algorithm [33]. We particularly derive the bounds of the sample size of the sub-sampled Riemannian Hessian and gradient, and show practical performance improvements of our algorithms over other Riemannian algorithms. We specifically address the case of compact submanifolds of $\mathbb{R}^n$ by following [33]. Additionally, the numerical experiments include problems on the Grassmann manifold to show effectiveness of our algorithms on more general quotient manifolds.

The paper is organized as follows. Section 2 describes the preliminaries and assumptions. We propose a novel inexact trust-region algorithm in the Riemannian setting in Section 3. In particular, in Section 4, we propose sub-sampled trust-region algorithms as its practical variants. Building upon the results in the Euclidean space [37, 38, 39] and that of the RTR algorithm [33], we derive the bounds of the sample size of sub-sampled gradients and Hessians in Theorem 4.1, which only requires a fixed sample size [37]. This has not been addressed in [37, 38, 39, 33]. In Section 5, numerical experiments on three different problems demonstrate significant speed-ups compared with state-of-the-art Riemannian deterministic and stochastic algorithms when the sample size $n$ is large.

The implementation of the proposed algorithms uses the MATLAB toolbox Manopt [40] and is available at `https://github.com/hiroyuki-kasai/Subsampled-RTR`. The proofs of theorems and additional experiments are provided as supplementary material.

## 2 Preliminaries and assumptions

We assume that $\mathcal{M}$ is endowed with a Riemannian metric structure, i.e., a smooth inner product $\langle \cdot, \cdot \rangle_x$ of tangent vectors is associated with the tangent space $T_x\mathcal{M}$ for all $x \in \mathcal{M}$. The *norm* $\| \cdot \|_x$ of a tangent vector in $T_x\mathcal{M}$ is the norm associated with the Riemannian metric. We also assume that $f$ is twice continuously differentiable throughout this paper.

### 2.1 Riemannian trust-region algorithm (RTR)

RTR is the generalization of the classical trust-region algorithm in the Euclidean space [41] to Riemannian manifolds [1, Chap. 7]. In comparison with the Euclidean case, in RTR, the *approximation model* $m_x$ of $f_x$ around $x$ is obtained from the Taylor expansion of the *pullback* of the function $\hat{f}_x \triangleq f_x \circ R_x$ defined on the tangent space, where $R_x$ is the retraction operator that maps a tangent vector onto the manifold with a local rigidity condition that preserves the gradients at $x$ [1, Chap. 4]. *Exponential mapping* is an instance of the retraction. $\hat{f}_x$ is a real-valued function on the *vector space* of $T_x\mathcal{M}$, and the pullback of $f_x$ at $x$ to $T_x\mathcal{M}$ through $R_x$, around the origin $0_x$ of $T_x\mathcal{M}$. This model of $m_x$ is denoted as $\hat{m}_x$, where $m_x = \hat{m}_x \circ R^{-1}$, and is chosen for $\xi \in T_x\mathcal{M}$ as

$$\hat{m}_x(\xi) \;=\; f(x) + \langle \mathrm{grad}f(x), \xi \rangle_x + \frac{1}{2} \langle H(x)[\xi], \xi \rangle_x, \tag{2}$$

where $H(x) : T_x\mathcal{M} \to T_x\mathcal{M}$ is some symmetric operator on $T_x\mathcal{M}$. The algorithm of RTR starts with an initial point $x_0 \in \mathcal{M}$, an initial radius $\Delta_0$, and a maximum radius $\Delta_{\max}$. At iteration $k$, RTR defines a *trust region* $\Delta_k$ around the current point $x_k \in \mathcal{M}$, which can be *trusted* such that it constructs a local model $\hat{m}_{x_k}$ that is a reasonable approximation of the the real objective function $\hat{f}_{x_k}$. It then finds the direction and the length of the step, denoted as $\eta_k$, simultaneously by solving a sub-problem based on the approximate model in this region. It should be noted that this calculation is performed in the vector space $T_{x_k}\mathcal{M}$. The next candidate iterate $x_k^+ = R_{x_k}(\eta_k)$ is accepted as $x_{k+1} = x_k^+$ when the decrease of the true objective function $\hat{f}_k(x_k) - \hat{f}_k(x_k^+)$ is sufficiently large against that of the approximate model $\hat{m}_k(0_{x_k}) - \hat{m}_k(\eta_k)$. Otherwise, we accept as $x_{k+1} = x_k$. Here, $\hat{f}_k$ and $\hat{m}_k$ represent $\hat{f}_{x_k}$ and $\hat{m}_{x_k}$, respectively, and hereinafter we use them for notational simplicity. The trust region $\Delta_k$ is enlarged, unchanged, or shrunk by the parameter $\gamma > 1$ according to the degree of the agreement of the model decrease and the true function decrease.

### 2.2 Essential assumptions

Since the first-order optimality condition, i.e., $\|\mathrm{grad}f(x)\|_x = 0$, is not sufficient in non-convex minimization problems due to existence of saddle points and local maximum points, we typically design algorithms that guarantee convergence to a point satisfying the second-order optimality conditions $\|\mathrm{grad}f(x)\|_x = 0$ and $\mathrm{Hess}f(x) \succeq 0$. In practice, however, we use its approximate condition, which is defined as $(\epsilon_g, \epsilon_H)$-optimality as presented below.

**Definition 2.1** (($\epsilon_g, \epsilon_H$)-optimality [42]). *Given $0 < \epsilon_g, \epsilon_H < 1$, $x$ is said to be an $(\epsilon_g, \epsilon_H)$-optimality of (1) when*

$$\|\mathrm{grad}f(x)\|_x \leq \epsilon_g, \quad \text{and} \quad \mathrm{Hess}f(x) \succeq -\epsilon_H \mathrm{Id},$$

*where $\mathrm{grad}f(x)$ is the Riemannian gradient, and $\mathrm{Hess}f(x)$ is the Riemannian Hessian of $f$ at $x$. Id is the identity mapping.*

We now provide essential assumptions below. We consider the inexact Hessian $H(x_k) : T_{x_k}\mathcal{M} \to T_{x_k}\mathcal{M}$ and the inexact gradient $G(x_k) \in T_{x_k}\mathcal{M}$ for $\mathrm{grad}f(x)$ in (2). Hereinafter, we particularly use $H_k \triangleq H(x_k)$ and $G_k \triangleq G(x_k)$ at $x_k$ for notational simplicity.

**Assumption 1** (Compact submanifold in $\mathbb{R}^n$ and second-order retraction). *We consider compact submanifolds in $\mathbb{R}^n$. We also assume that the retraction is the second-order retraction.*

It should be noted that, although the Hessian $\nabla^2 \hat{f}_x(0_x)$ and the Riemannian Hessian $\mathrm{Hess}f(x)$ are in general different from each other, they are *identical* under *second-order* retraction [33, Lem. 17]. This assumption ensures that, as stated in Theorem 3.1, Algorithm 1 provides a solution that satisfies the $(\epsilon_g, \epsilon_H)$-optimality. Otherwise, it gives a solution satisfying $\lambda_{\min}(H(x)) \geq -\epsilon_H$. It should be stressed that the second-order retractions are available in many submanifolds such as $R_x(\eta) = (x+\eta)/\|x+\eta\|_x$ in the case of spherical manifold [1, Sec. 4].

**Assumption 2** (Restricted Lipschitz Hessian [33, A.5]). *If $\epsilon_H < \infty$, there exists $L_H \geq 0$ such that, for all $x_k$, $\hat{f}_k$ satisfies*

$$\left| \hat{f}_k(\eta_k) - f(x_k) - \langle \mathrm{grad} f(x_k), \eta_k \rangle_{x_k} - \frac{1}{2} \langle \eta_k, \nabla^2 \hat{f}_k(0_{x_k})[\eta_k] \rangle_{x_k} \right| \quad \leq \quad \frac{1}{2} L_H \|\eta_k\|_{x_k}^3,$$

*for all $\eta_k \in T_{x_k}\mathcal{M}$ such that $\|\eta\|_{x_k} \leq \Delta_k$.*

It should be noted that the retraction $R_x$ needs to be defined *only* in the radius of $\Delta_k$. Since the manifold under consideration is compact, Assumption 2 holds [33, Lem. 9]. We also assume a bound of the norm of the inexact Riemannian Hessian $H_k$ [33, A.6].

**Assumption 3** (Norm bound on $H_k$). *There exists $K_H \geq 0$ such that, for all $x_k$, $H_k$ satisfies*

$$\|H_k\|_{x_k} \quad \triangleq \quad \sup_{\eta \in T_{x_k}\mathcal{M}, \|\eta\|_{x_k} \leq 1} \langle \eta, H_k[\eta] \rangle_{x_k} \leq K_H.$$

We now provide essential assumptions on the bounds for approximation error of the inexact Riemannian gradient $G_k$ and the inexact Riemannian Hessian $H_k$ at iteration $k$. As seen later in Section 4, this ensures that the sample size of sub-sampling can be fixed.

**Assumption 4** (Approximation error bounds on inexact gradient and Hessian). *There exist constants $0 < \delta_g, \delta_H < 1$ such that the approximation of the gradient, $G_k$, and the approximation of the Hessian, $H_k$, at iterate $k$, satisfy*

$$\|G_k - \mathrm{grad} f(x_k)\|_{x_k} \quad \leq \quad \delta_g, \tag{3}$$
$$\|(H_k - \nabla^2 \hat{f}_k(0_{x_k}))[\eta_k]\|_{x_k} \quad \leq \quad \delta_H \|\eta_k\|_{x_k}. \tag{4}$$

The latter is a *weaker* condition than the below condition [33, A7].

$$\|H_k - \nabla^2 \hat{f}_k(0_{x_k})\|_{x_k} \quad \leq \quad \delta_H.$$

It should be emphasized that the approximation error bound for $H_k$ is defined with the Hessian of the pullback of $f$ at $x_k$, i.e., $\nabla^2 \hat{f}_k(0_{x_k})$, instead of the Riemannian Hessian of $f$, i.e., $\mathrm{Hess} f(x_k)$. Furthermore, it should be noted that Assumption 4 is a *relax* form in comparison with a typical condition in the Euclidean setting, which is defined as [43, AM.4]

$$\|(H_k - \nabla^2 \hat{f}_k(0_{x_k}))[\eta_k]\|_{x_k} \quad \leq \quad \delta_H \|\eta_k\|_{x_k}^2. \tag{5}$$

This typical form (5) is different from (4). It should be noted that the condition (5) requires that the sizes of the sub-sampled Hessian and gradient need to be *increased* towards the convergence, whereas our new condition (4) allows the size to be *fixed*, as seen later in Section 4 [37, 38].

Finally, we give an assumption for the step $\eta_k$. We need a sufficient decrease in $\hat{m}_k(\eta_k)$, and there exit ways to solve the sub-problem (See [41, 1] for more details). However, the calculation of the exact solution of the problem is prohibitive, especially in large-scale problems. To this end, various approximate solvers have been investigated in the literature that require certain conditions to be met. The popular conditions are the Cauchy and Eigenpoint conditions [41]. The assumptions required for the convergence analysis of Algorithm 1 by generalizing [37, Cond. 2] are provided below.

**Assumption 5** (Sufficient descent relative to the Cauchy and Eigen directions [41, 37]). *We assume the first-order step, called the Cauchy step, as*

$$\hat{m}_k(0_{x_k}) - \hat{m}_k(\eta_k) \quad \geq \quad \hat{m}_k(0_{x_k}) - \hat{m}_k(\eta_k^C) \quad \geq \quad \frac{1}{2}\|G_k\|_{x_k} \min\left\{ \frac{\|G_k\|_{x_k}}{1 + \|H_k\|}, \Delta_k \right\}.$$

*We assume the second-order step, called the Eigen step, for some $\nu \in (0, 1]$ when $\lambda_{\min}(H_k) < -\epsilon_H$ as*

$$\hat{m}_k(0_{x_k}) - \hat{m}_k(\eta_k) \quad \geq \quad \hat{m}_k(0_{x_k}) - \hat{m}_k(\eta_k^E) \quad \geq \quad \frac{1}{2}\nu|\lambda_{\min}(H_k)|\Delta_k^2.$$

Here, $\eta_k^C$ is the negative gradient direction and $\eta_k^E$ is an approximation of the negative curvature direction such that $\langle \eta_k^E, H_k[\eta_k^E] \rangle_{x_k} \leq \nu \lambda_{\min}(H_k)\|\eta_k^E\|_{x_k}^2 < 0$. Assumption 5 is ensured by using TR subproblem solvers, e.g., the Steihaug-Toint truncated conjugate gradients algorithm [44].

---

**Algorithm 1** Inexact Riemannian trust-region (Inexact RTR) algorithm

---

**Require:** $0 < \Delta_{\max} < \infty$, $\epsilon_g, \epsilon_H \in (0,1)$, $\rho_{TH}, \gamma > 1$.
 1: Initialize $0 < \Delta_0 < \Delta_{\max}$, and a starting point $x_0 \in \mathcal{M}$.
 2: **for** $k = 1, 2, \ldots$ **do**
 3:    Set the approximate (inexact) gradient $G_k$ and $H_k$.
 4:    **if** $\|G_k\| \leq \epsilon_g$ and $\lambda_{\min}(H_k) \geq -\epsilon_H$ **then** Return $x_k$. **end if**
 5:    **if** $\|G_k\| \leq \epsilon_g$ **then** $G_k = 0$. **end if**
 6:    Calculate $\eta_k \in T_{x_k}\mathcal{M}$ by solving $\eta_k \approx \underset{\|\eta\| \leq \Delta_k}{\arg\min} f(x_k) + \langle G_k, \eta \rangle_{x_k} + \frac{1}{2}\langle \eta, H_k[\eta]\rangle_{x_k}$.
 7:    Set $\rho_k = \frac{\hat{f}_k(0_{x_k}) - \hat{f}_k(\eta_k)}{\hat{m}_k(0_{x_k}) - \hat{m}_k(\eta_k)}$.
 8:    **if** $\rho_k \geq \rho_{TH}$ **then** $x_{k+1} = R_{x_k}(\eta_k)$ and $\Delta_{k+1} = \gamma\Delta_k$.
 9:    **else** $x_{k+1} = x_k$ and $\Delta_{k+1} = \Delta_k/\gamma$. **end if**
10: **end for**
11: Output $x_k$.

---

## 3 Riemannian trust-regions with inexact Hessian and gradient

This section proposes an inexact variant of the Riemannian trust-region algorithm, i.e., inexact RTR, which approximates gradient and Hessian as well as the solution of a sub-problem. The proposed algorithm is summarized in Algorithm 1. The inexact RTR algorithm solves approximately a sub-problem $\hat{m}_k(\eta) : T_{x_k}\mathcal{M} \to \mathbb{R}$ for $\eta \in T_{x_k}\mathcal{M}$ of the form

$$\eta_k \quad \approx \quad \underset{\eta \in T_{x_k}\mathcal{M}}{\arg\min} \ \hat{m}_k(\eta) \qquad \text{subject to} \qquad \|\eta\|_{x_k} \leq \Delta_k, \tag{6}$$

where $\hat{m}_k(\eta)$ is notably defined as

$$\hat{m}_k(\eta) = \begin{cases} f(x_k) + \langle G_k, \eta \rangle_{x_k} + \dfrac{1}{2}\langle \eta, H_k[\eta]\rangle_{x_k}, & \|G_k\|_{x_k} \geq \epsilon_g, \tag{7a} \\[2mm] f(x_k) + \dfrac{1}{2}\langle \eta, H_k[\eta]\rangle_{x_k}, & \text{otherwise.} \tag{7b} \end{cases}$$

It should be stressed that, as (7b) represents, we ignore the gradient when it is smaller than $\epsilon_g$, i.e., $\|G_k\|_{x_k} < \epsilon_g$, which is crucial for the convergence analysis in Theorem 3.1 [38].

Now, we show the convergence analysis of the proposed inexact RTR. To this end, we assume an additional approximation condition on the inexact gradient and Hessian for the constants in Assumption 4 [38, Cond. 1]. This additional assumption is essential for the relax form of (4).

**Assumption 6** (Gradient and Hessian approximations for Algorithm 1 [38]). *Let $\rho_{TH}$ be the threshold parameter of the reduction ratio of the true objective function and the approximate model in Algorithm 1. For $\nu \in (0,1]$ in Assumption 5, we assume that the constants of the inexact gradient and Hessian satisfy $\delta_g < \frac{1-\rho_{TH}}{4}\epsilon_g$ and $\delta_H < \min\left\{\frac{1-\rho_{TH}}{2}\nu\epsilon_H, 1\right\}$.*

This implies that we only need $\delta_g \in \mathcal{O}(\epsilon_g)$ and $\delta_H \in \mathcal{O}(\epsilon_H)$ [38, Cond. 1].

**Theorem 3.1** (Optimal complexity of Algorithm 1). *Consider $0 < \epsilon_g, \epsilon_H < 1$. Suppose Assumptions 1, 2, and 3 hold. Also, suppose that the inexact Hessian $H_k$ and gradient $G_k$ satisfy Assumption 4 with the approximation tolerance $\delta_g$ and $\delta_H$. Suppose that the solution of the sub-problem (6) satisfies Assumption 5 and Assumption 6 holds. Then, Algorithm 1 returns an $(\epsilon_g, \epsilon_H)$-optimal solution in, at most, $T \in \mathcal{O}(\max\{\epsilon_g^{-2}\epsilon_H^{-1}, \epsilon_H^{-3}\})$ iterations.*

The proof of Theorem 3.1 follows that of [37, 38, 33]. Therefore, we only provide the proof sketch in Section B.1 of the supplementary material file.

## 4 Sub-sampled Riemannian trust-regions for finite-sum problems

Particularly addressing large-scale finite-sum minimization problems, we propose an inexact gradient and Hessian trust-region algorithm, Sub-RTR, by exploiting a sub-sampling technique to generate inexact gradient and Hessian. The generated inexact gradient and Hessian satisfy Assumption 4 in a *probabilistic way*. More concretely, we derive sampling conditions based on the probabilistic

deviation bounds for random matrices, which originate from the *Bernstein inequality* in Lemma B.2 of the supplementary material file.

We first define the sub-sampled inexact gradient and Hessian as

$$G_k \triangleq \frac{1}{|\mathcal{S}_g|} \sum_{i \in \mathcal{S}_g} \mathrm{grad} f_i(x_k) \quad \text{and} \quad H_k \triangleq \frac{1}{|\mathcal{S}_H|} \sum_{i \in \mathcal{S}_H} \mathrm{Hess} f_i(x_k), \qquad i = 1, 2, \ldots, n,$$

where $\mathcal{S}_g, \mathcal{S}_H \subset \{1, \ldots, n\}$ are the set of the sub-sampled indexes for the estimates of the approximate gradient and Hessian, respectively. Their sizes, i.e., the cardinalities, are denoted as $|\mathcal{S}_g|$ and $|\mathcal{S}_H|$, respectively. Next, we provide the sampling conditions. For simplicity, we use the standard Riemannian metric in the analysis. Equivalently, $\mathcal{M}$ is endowed with a smooth inner product $\langle \cdot, \cdot \rangle_2$ and the norm $\| \cdot \|_2$. We suppose that

$$\sup_{x \in \mathcal{M}} \|\mathrm{grad} f_i(x)\|_2 \leq K_g^i \quad \text{and} \quad \sup_{x \in \mathcal{M}} \|\mathrm{Hess} f_i(x)\|_2 \leq K_H^i \qquad i = 1, 2, \ldots, n,$$

and we also define $K_g^{\max} \triangleq \max_i K_g^i$ and $K_H^{\max} \triangleq \max_i K_H^i$. As for the sufficient size of sub-sampling to guarantee the convergence in Theorem 3.1, we have the following theorem.

**Theorem 4.1** (Bounds on sampling size). *Given $K_g^i, K_g^{\max}$ and $K_H^i, K_H^{\max}$, and $0 < \delta, \delta_g, \delta_H < 1$, we define*

$$|\mathcal{S}_g| \geq \frac{32(K_g^{\max})^2 \log(1/\delta) + 1/4}{\delta_g^2} \quad \text{and} \quad |\mathcal{S}_H| \geq \frac{32(K_H^{\max})^2 \log(1/\delta) + 1/4}{\delta_H^2}.$$

*At any $x_k \in \mathcal{M}$, suppose that the sampling is done uniformly at random to generate $\mathcal{S}_g$ and $\mathcal{S}_H$. Then, we have*

$$\mathrm{Pr}(\|G_k - \mathrm{grad} f(x_k)\|_2 \leq \delta_g) \geq 1 - \delta,$$
$$\mathrm{Pr}(\|(H_k - \nabla^2 \hat{f}_k(0_x))[\eta_k]\|_2 \leq \delta_H \|\eta_k\|_2) \geq 1 - \delta.$$

From Theorem 4.1, it can be easily seen that Assumption 4 follows with the same probability with $K_g = K_g^{\max}$ and $K_H = K_H^{\max}$. It should be emphasized that if we use the typical condition (5) instead of Assumption 4, we obtain, e.g., $|\mathcal{S}_H| \geq \frac{32(K_H^{\max})^2 \log(1/\delta) + 1/4}{\delta_H^2 \|\eta_k\|_2^2}$ for the sub-sampled Hessian $H_k$. Considering that $\|\eta_k\|$ goes to nearly zero as the iterations proceed, this obtained bound indicates that $|\mathcal{S}_H|$ increases accordingly. Consequently, the size of the sub-sampled Hessian needs to be increased towards the convergence. On the other hand, our results ensure that the sample size can be fixed to guarantee the convergence of Algorithm 1.

## 5 Numerical comparisons

This section evaluates the performance of our two proposed inexact RTR algorithms: the sub-sampled Hessian RTR (Sub-H-RTR) and the sub-sampled Hessian and gradient RTR (Sub-HG-RTR). We compare them with the Riemannian deterministic algorithms: RSD, Riemannian conjugate gradient (RCG), RLBFGS, and RTR. We also show comparisons with RSVRG [20, 21]. We compare the algorithms in terms of the total number of *oracle calls* and run time, i.e., "wall-clock" time. The former measures the number of function, gradient, and Hessian-vector product computations. The sub-sampled RTR requires $(n + |\mathcal{S}_g| + r_s |\mathcal{S}_H|)$ oracle calls per iteration, whereas the original RTR requires $(2n + r_s n)$ oracle calls. Here, $r_s$ is the number of iterations required for solving the trust-region sub-problem approximately. RSD, RCG, and RLBFGS require $(n + r_l n)$ oracle calls per iteration, where $r_l$ is the number of line searches carried out. RSVRG requires $(n + mn)$ oracle calls per *outer* iteration, where $m$ is the update frequency of the outer loop. Algorithms are initialized randomly and are stopped when either the gradient norm is below a particular threshold. Multiple constant stepsizes from $\{10^{-10}, 10^{-9}, \ldots, 1\}$ are used for RSVRG and the best-tuned results are shown. By following [38], we set $|\mathcal{S}_g| = n/10$ and $|\mathcal{S}_H| = n/10^2$ except **Cases P5**, **P6**, **M4**, and **M5**. We set the batch-size to $n/10$ in RSVRG. All simulations are performed in MATLAB on a 4.0 GHz Intel Core i7 machine with 32 GB RAM.

We address the independent component analysis (ICA) problem on the *Stiefel* manifold and two problems on the *Grassmann* manifold, namely the principal component analysis (PCA) and the low-rank matrix completion (MC) problems. The Stiefel manifold is the set of orthogonal $r$-frames in

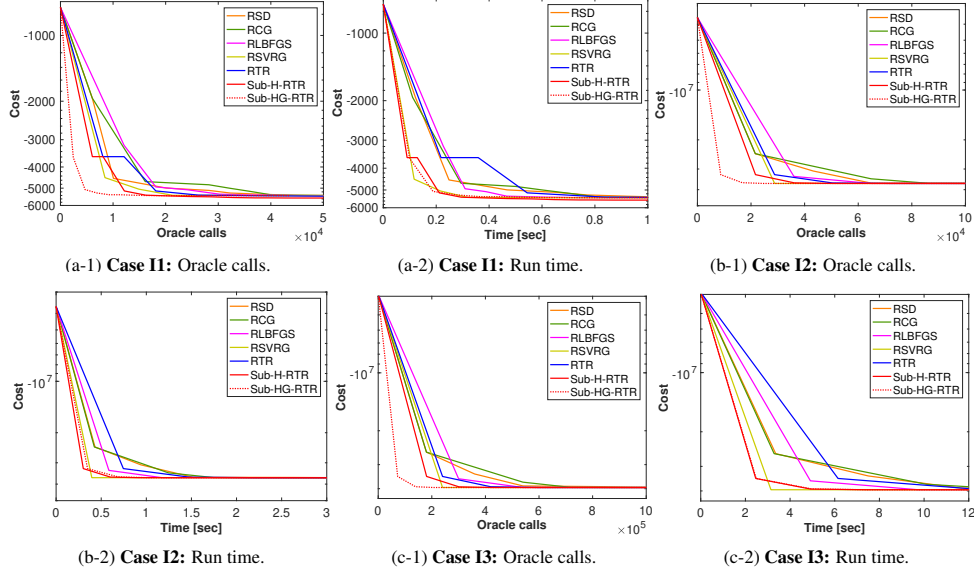

(a-1) **Case I1:** Oracle calls.     (a-2) **Case I1:** Run time.     (b-1) **Case I2:** Oracle calls.

(b-2) **Case I2:** Run time.     (c-1) **Case I3:** Oracle calls.     (c-2) **Case I3:** Run time.

Figure 1: Performance evaluations on the ICA problem.

$\mathbb{R}^d$ for some $r \leq d$ and is viewed as an embedded submanifold of $\mathbb{R}^{d \times r}$ [1, Sec. 3.3]. On the other hand, the Grassmann manifold $\mathrm{Gr}(r, d)$ is the set of $r$-dimensional subspaces in $\mathbb{R}^d$ and is a Riemannian quotient manifold of the Stiefel manifold [1, Sec. 3.4]. The motivation behind including the latter two applications is to show that our proposed algorithms empirically work very well even if the manifold is not a submanifold. In all these problems, full gradient methods, i.e., RSD, RCG, RLBFGS, and RTR, become prohibitively computationally expensive when $n$ is very large and the inexact approach is one promising way to achieve scalability. The details of the manifolds and the derivations of the Riemannian gradient and Hessian are provided as supplementary material.

## 5.1 ICA problem

The ICA or the blind source separation problem refers to separating a signal into components so that the components are as independent as possible [45]. A particular preprocessing step is the whitening step that is proposed through joint diagonalization on the Stiefel manifold [13], i.e., $\min_{\mathbf{U} \in \mathbb{R}^{d \times r}} -\frac{1}{n} \sum_{i=1}^{n} \|\mathrm{diag}(\mathbf{U}^\top \mathbf{C}_i \mathbf{U})\|_F^2$, where $\|\mathrm{diag}(\mathbf{A})\|_F^2$ defines the sum of the squared diagonal elements of $\mathbf{A}$. The symmetric matrices $\mathbf{C}_i$s are of size $d \times d$ and can be cumulant matrices or time-lagged covariance matrices of different signal samples [13].

We use three real-world datasets: `YaleB` [46], `COIL-100` [47], and `CIFAR-100` [48]. From these datasets, we create a Gabor-Based region covariance matrix (GRCM) descriptor [49, 50, 51]. A $43 \times 43$ GRCM is computed from the pixel coordinates and Gabor features that are obtained by convolving Gabor kernels with an intensity image. We set $m = 1$ in RSVRG. Figures 1 (a), (b), and (c) show the results on the `YaleB` dataset with $(n, d, r) = (2015, 43, 43)$ (**Case I1**), the `COIL-100` dataset with $(n, d, r) = (7.2 \times 10^3, 43, 43)$ (**Case I2**) and the `CIFAR-100` dataset with $(n, d, r) = (6 \times 10^4, 43, 43)$ (**Case I3**), respectively. As seen, the proposed Sub-H-RTR and Sub-HG-RTR perform better in terms of both the number of oracle calls and run time than others except RSVRG. It should be emphasized that though RSVRG performs comparable to or slightly better than our proposed algorithms, its results require *fine tuning* of stepsizes.

## 5.2 PCA problem

Given an orthonormal matrix projector $\mathbf{U} \in \mathrm{St}(r, d)$, the PCA problem is to minimize the sum of squared residual errors between projected data points and the original data as $\min_{\mathbf{U} \in \mathrm{St}(r,d)} \frac{1}{n} \sum_{i=1}^{n} \|\mathbf{z}_i - \mathbf{U}\mathbf{U}^\top \mathbf{z}_i\|_2^2$, where $\mathbf{z}_i$ is a data vector of size $d \times 1$. This problem is equivalent to $\min_{\mathbf{U} \in \mathrm{St}(r,d)} = -\frac{1}{n} \sum_{i=1}^{n} \mathbf{z}_i^\top \mathbf{U}\mathbf{U}^\top \mathbf{z}_i$. Here, the critical points in the space $\mathrm{St}(r, d)$ are not isolated because the cost function remains unchanged under the group action $\mathbf{U} \mapsto \mathbf{U}\mathbf{O}$ for all orthogonal matrices $\mathbf{O}$ of size $r \times r$. Subsequently, the PCA problem is an optimization problem on the Grassmann manifold $\mathrm{Gr}(r, d)$.

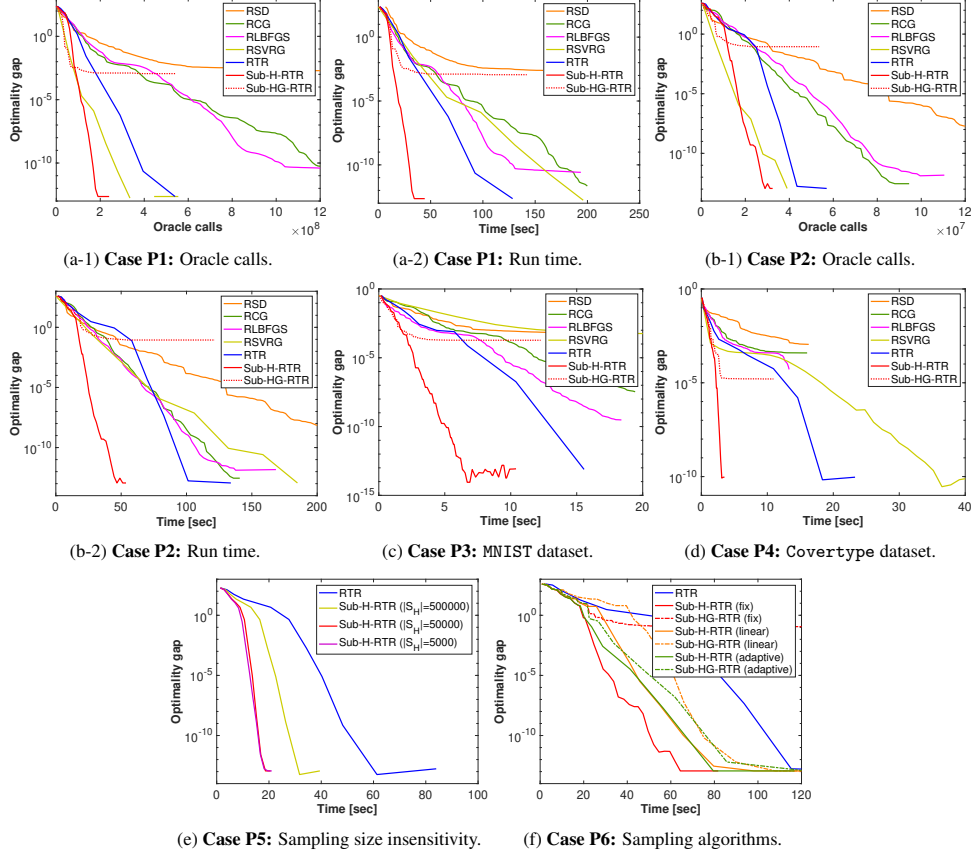

(a-1) **Case P1:** Oracle calls.     (a-2) **Case P1:** Run time.     (b-1) **Case P2:** Oracle calls.

(b-2) **Case P2:** Run time.     (c) **Case P3:** `MNIST` dataset.     (d) **Case P4:** `Covertype` dataset.

(e) **Case P5:** Sampling size insensitivity.     (f) **Case P6:** Sampling algorithms.

Figure 2: Performance evaluations on the PCA problem.

Figures 2(a) and (b) show the results on two synthetic datasets with $(n, d, r) = (5 \times 10^6, 10^2, 5)$ (**Case P1**), and $(n, d, r) = (5 \times 10^5, 10^3, 5)$ (**Case P2**). We set $m = 5$ in RSVRG. It should be noted that, although RSVRG is competitive in terms of the oracle calls in (a), its run time performance is poor than others. This is attributed to RSVRG requiring retraction and vector transport operations at every iteration. Overall, the proposed Sub-H-RTR outperforms others, whereas the proposed Sub-HG-RTR is inferior to others. Figures 2(c) and (d) show the results on two real-world datasets with $r = 10$, where **Case P3** deals with the `MNIST` dataset [52] with $(n, d) = (6 \times 10^4, 784)$ and **Case P4** deals with the `Covertype` dataset [53] with $(n, d) = (581012, 54)$. From the figure, our proposed Sub-H-RTR outperforms others. We also change the sample size in Sub-H-RTR as $|\mathcal{S}_H| = \{n/10, n/10^2, n/10^3\}$ in **Case P1**. We observe that Sub-H-RTR has low sensitivity to the size $|\mathcal{S}_H|$ from Figures 2(e) (**Case P5**). Additionally, we compare three different ways to decide the sample size of $|\mathcal{S}_H|$ and $|\mathcal{S}_g|$: (i) "fixed", (ii) "linear", and (iii) "adaptive" variants (**Case P6**). The "fixed" variant keeps the size as the initial $|\mathcal{S}_g|$ and $|\mathcal{S}_H|$ as theoretically supported by Theorem 4.1. The "linear" variant uses $k|\mathcal{S}_g|$ and $k|\mathcal{S}_H|$ at iteration $k$. The "adaptive" variant decides the sizes based on (5) [39]. The results on the synthetic dataset same as **Case P2** show that all the proposed algorithms except Sub-HG-RTR with fixed sample size outperform the original RTR.

## 5.3 MC problem

The MC problem amounts to completing an incomplete matrix $\mathbf{Z}$, say of size $d \times n$, from a small number of entries by assuming a low-rank model for the matrix. If $\Omega$ is the set of the indices for which we know the entries in $\mathbf{Z}$, the rank-$r$ MC problem amounts to solving the problem $\min_{\mathbf{U} \in \mathbb{R}^{d \times r}, \mathbf{A} \in \mathbb{R}^{r \times n}} \|\mathcal{P}_\Omega(\mathbf{UA}) - \mathcal{P}_\Omega(\mathbf{Z})\|_F^2$, where the operator $\mathcal{P}_\Omega(\mathbf{Z}_{pq}) = \mathbf{Z}_{pq}$ if $(p, q) \in \Omega$ and $\mathcal{P}_\Omega(\mathbf{Z}_{pq}) = 0$ otherwise is called the orthogonal sampling operator and is a mathematically convenient way to represent the subset of known entries. Partitioning $\mathbf{Z} = [\boldsymbol{z}_1, \boldsymbol{z}_2, \ldots, \boldsymbol{z}_n]$, the problem is equivalent to the problem $\min_{\mathbf{U} \in \mathbb{R}^{d \times r}, \boldsymbol{a}_i \in \mathbb{R}^r} \frac{1}{n} \sum_{i=1}^n \|\mathcal{P}_{\Omega_i}(\mathbf{U}\boldsymbol{a}_i) - \mathcal{P}_{\Omega_i}(\boldsymbol{z}_i)\|_2^2$, where $\boldsymbol{z}_i \in \mathbb{R}^d$ and the operator $\mathcal{P}_{\Omega_i}$ is the sampling operator for the $i$-th column.

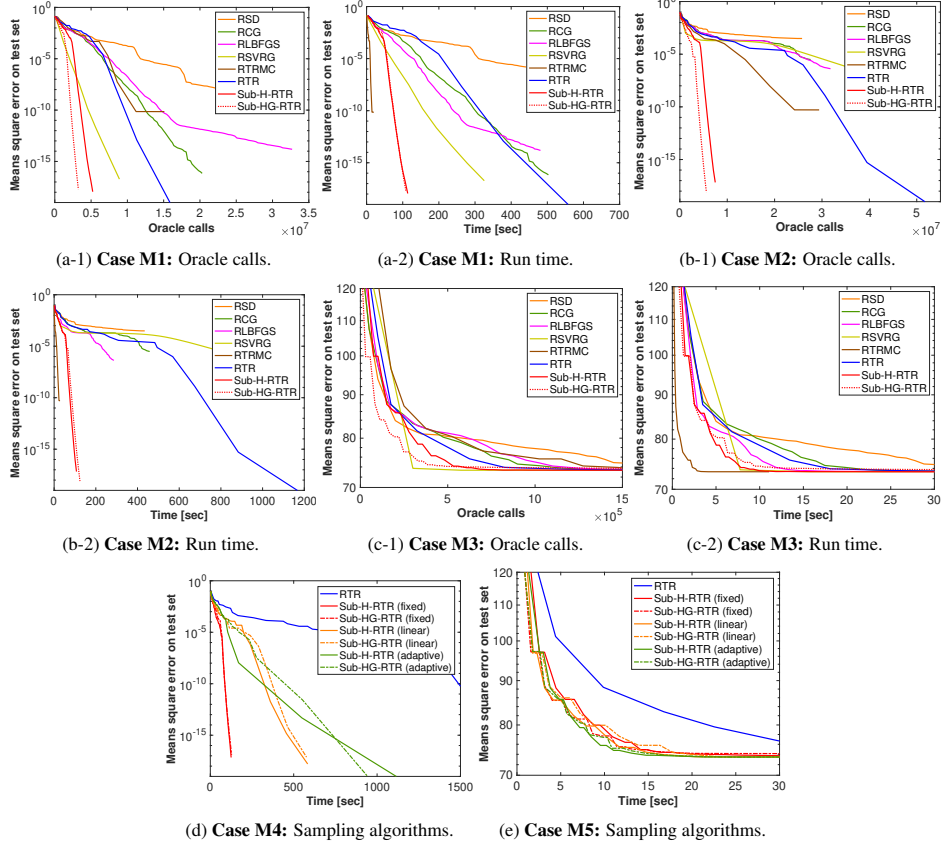

(a-1) **Case M1:** Oracle calls.     (a-2) **Case M1:** Run time.     (b-1) **Case M2:** Oracle calls.

(b-2) **Case M2:** Run time.     (c-1) **Case M3:** Oracle calls.     (c-2) **Case M3:** Run time.

(d) **Case M4:** Sampling algorithms.     (e) **Case M5:** Sampling algorithms.

Figure 3: Performance evaluations on the MC problem.

We also compared our proposed algorithms with RTRMC [10], a state-of-the-art MC algorithm. The code of RTRMC is optimized for the MC problem. Therefore, we mainly compare the oracle calls of RTRMC for fair comparison. We first consider a synthetic dataset with $(n, d, r) = (10^5, 10^2, 5)$. We show the mean squares error (MSE) on a *test set*, which is different from the *training set*. The over-sampling ratio (OS) is 4, where the OS determines the number of entries that are known. An OS of 4 implies that $4(n + d - r)r$ number of randomly and uniformly selected entries are known a priori out of the total $nd$ entries. We also impose an *exponential decay* of singular values. The ratio of the largest to the lowest singular value is known as the condition number (CN) of the matrix. We set $m = 5$ in RSVRG. We consider a well-conditioned case with CN=5 (**Case M1**) and an ill-conditioned case with CN=20 (**Case M2**). Figures 3(a) and (b) show relatively good performance of RSVRG for **Case M1** . RTRMC is, as expected, extremely fast in terms of run time (owing to its optimized code). Sub-H-RTR and Sub-HG-RTR show superior performance than others, especially for the ill-conditioned case **M2**. Next, we consider the `Jester` dataset 1 [54] consisting of ratings of 100 jokes by 24983 users (**Case M3**). Each rating is a real number between $-10$ and $10$. The algorithms are run by fixing the rank to $r = 5$. Figure 3(c) shows the comparable or superior performance of the sub-sampled RTR on the test sets against state-of-the-art algorithms. Finally, we compare three variants: "fixed", "linear", and "adaptive" to decide the sample size in **Cases M4** and **M5** under the same conditions as **Cases M2** and **M3**, respectively. Figures 3(d) and (e) show that all the proposed algorithms outperform the original RTR. In particular, the "fixed" variant gives superior performance than others as supported by Theorem 4.1.

## 6   Conclusion

We have proposed an inexact trust-region algorithm in the Riemannian setting with a worst case total complexity bound. Additionally, we have also proposed sub-sampled trust-region algorithms for finite-sum problems, which need only fixed sample bounds of sub-sampled gradient and Hessian. The numerical comparisons show the benefits of our proposed inexact RTR algorithms on a number of applications.

## Acknowledgements

H. Kasai was partially supported by JSPS KAKENHI Grant Numbers JP16K00031 and JP17H01732. We thank Nicolas Boumal and Hiroyuki Sato for insight discussions and also express our sincere appreciation to Jonas Moritz Kohler for sharing his expertise on sub-sampled algorithms in the Euclidean case.

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
