[Supplementary Material · neurips_2018_InexactTR_supplementary.pdf]

# Supplementary material for
# Inexact trust-region algorithms on
# Riemannian manifolds

**Hiroyuki Kasai**
The University of Electro-Communications
Japan
kasai@is.uec.ac.jp

**Bamdev Mishra**
Microsoft
India
bamdevm@microsoft.com

## Abstract

This supplementary file presents the overview of the manifolds of interest, the proof of the convergence analysis, and additional numerical experiments.

## A  Manifolds and problems of interest

### A.1  Manifolds

**Stiefel manifold** $\mathrm{St}(r,d)$**:** The Stiefel manifold is the set of orthogonal $r$-frames in $\mathbb{R}^d$ for some $r \leq d$, and it is an embedded submanifold of $\mathbb{R}^{d\times r}$. The orthogonal group $\mathrm{O}(d)$ is a special case of the Stiefel manifold, i.e., $\mathrm{O}(d) = \mathrm{St}(d,d)$. Because $\mathrm{St}(r,d)$ is a submanifold embedded in $\mathbb{R}^{d\times r}$, we can endow the canonical inner product in $\mathbb{R}^{d\times r}$ as a Riemannian metric $\langle \xi, \eta \rangle_{\mathbf{U}} = \mathrm{tr}(\xi^\top \eta)$ for $\xi, \eta \in T_{\mathbf{U}}\mathrm{St}(r,d)$. With this Riemannian metric, the projection onto the tangent space $T_{\mathbf{U}}\mathrm{St}(r,d)$ is defined as an orthogonal projection $\mathrm{P}_{\mathbf{U}}(\mathbf{W}) = \mathbf{W} - \mathbf{U}\mathrm{sym}(\mathbf{U}^\top \mathbf{W})$ for $\mathbf{U} \in \mathrm{St}(r,d)$ and $\mathbf{W} \in \mathbb{R}^{d\times r}$. A popular retraction is $R_{\mathbf{U}}(\xi) = \mathrm{qf}(\mathbf{U}+\xi)$ for $\mathbf{U} \in \mathrm{St}(r,d)$ and $\xi \in T_{\mathbf{U}}\mathrm{St}(r,d)$, where $\mathrm{qf}(\cdot)$ extracts the orthonormal factor based on QR decomposition. Other details about optimization-related notions on the Stiefel manifold are in [1].

**Grassmann manifold** $\mathrm{Gr}(r,d)$**:**  A point on the Grassmann manifold is an equivalence class represented by a $d \times r$ orthogonal matrix $\mathbf{U}$ with orthonormal columns, i.e., $\mathbf{U}^\top \mathbf{U} = \mathbf{I}$. Two orthogonal matrices express the same element on the Grassmann manifold if they are related by right multiplication of an $r \times r$ orthogonal matrix $\mathbf{O} \in \mathrm{O}(r)$. Equivalently, an element of $\mathrm{Gr}(r,d)$ is identified with a set of $d \times r$ orthogonal matrices $[\mathbf{U}] := \{\mathbf{U}\mathbf{O} : \mathbf{O} \in \mathrm{O}(r)\}$. That is, $\mathrm{Gr}(r,d) := \mathrm{St}(r,d)/\mathrm{O}(r)$, where $\mathrm{St}(r,d)$ is the *Stiefel manifold* that is the set of matrices of size $d \times r$ with orthonormal columns. The Grassmann manifold has the structure of a Riemannian quotient manifold [1]. A popular retraction on the Grassmann manifold is $R_{\mathbf{U}}(\xi) = \mathrm{qf}(\mathbf{U}+\xi)$. Other details about optimization-related notions on the Grassmann manifold are in [1].

### A.2  Problems and derivations of Riemannian gradient and Hessian

**ICA problem** [12, 13]: A particular variant to solve the independent components analysis (ICA) problem is through joint diagonalization on the Stiefel manifold, i.e.,

$$\min_{\mathbf{U}\in\mathbb{R}^{d\times r}} f_{\mathrm{ica}}(\mathbf{U}) := -\frac{1}{n}\sum_{i=1}^{n} \|\mathrm{diag}(\mathbf{U}^\top \mathbf{C}_i \mathbf{U})\|_F^2,$$

where $\|\mathrm{diag}(\mathbf{A})\|_F^2$ defines the sum of the squared diagonal elements of $\mathbf{A}$. $\mathbf{C}_i$ can, for example, be cumulant matrices or time-lagged covariance matrices of size $d \times d$. The Riemannian gradient

$\operatorname{grad} f_{\text{ica}}(\mathbf{U})$ of the cost function $f_{\text{ica}}(\mathbf{U})$ is

$$\operatorname{grad} f_{\text{ica}}(\mathbf{U}) = \mathrm{P}_{\mathbf{U}} \operatorname{egrad} f_{\text{ica}}(\mathbf{U}) = \mathrm{P}_{\mathbf{U}}\left(-\frac{1}{n}\sum_{i=1}^{n} 4\mathbf{C}_i \mathbf{U}\operatorname{ddiag}(\mathbf{U}^{\top}\mathbf{C}_i\mathbf{U})\right),$$

where $\operatorname{egrad} f_{\text{ica}}(\mathbf{U})$ is the Euclidean gradient of $f_{\text{ica}}(\mathbf{U})$, ddiag is the diagonal matrix, and $\mathrm{P}_{\mathbf{U}}$ denotes the orthogonal projection onto the tangent space of $\mathbf{U}$, i.e., $T_{\mathbf{U}}\mathrm{St}(r,d)$, which is defined as $\mathrm{P}_{\mathbf{U}}(\mathbf{W}) = \mathbf{W} - \mathbf{U}\operatorname{sym}(\mathbf{U}^{\top}\mathbf{W})$, where $\operatorname{sym}(\mathbf{A})$ represents the symmetric matrix $(\mathbf{A}+\mathbf{A}^{\top})/2$. The Riemannian Hessian of $f_{\text{ica}}(\mathbf{U})$ along a search direction $\xi \in T_{\mathbf{U}}\mathrm{St}(r,d)$ is $\operatorname{Hess} f_{\text{ica}}(\mathbf{U})[\xi] = \nabla_{\xi}\operatorname{grad} f_{\text{ica}}(\mathbf{U})$, where $\nabla_{\xi}$ represents the Riemannian connection on $\mathcal{M}$. For the case of interest, $\nabla_{\eta}\xi = \mathrm{P}_{\mathbf{U}}(\mathrm{D}\xi(\mathbf{Y})[\eta])$, where $\mathbf{Y}$ represents the roof of $\eta \in T_{\mathbf{Y}}\mathcal{M}$. Consequently, the Riemannian Hessian is defined by

$$\begin{aligned}
\operatorname{Hess} f_{\text{ica}}(\mathbf{U})[\xi] = \quad & \mathrm{P}_{\mathbf{U}}\Big(\operatorname{Degrad} f_{\text{ica}}(\mathbf{U})[\xi] - \xi\operatorname{sym}(\mathbf{U}^{\top}\operatorname{egrad} f_{\text{ica}}(\mathbf{U})) \\
& - \mathbf{U}\operatorname{sym}(\xi^{\top}\operatorname{egrad} f_{\text{ica}}(\mathbf{U})) - \mathbf{U}\operatorname{sym}(\mathbf{U}^{\top}\operatorname{Degrad} f_{\text{ica}}(\mathbf{U})[\xi])\Big).
\end{aligned}$$

Here, $\operatorname{Degrad} f_{\text{ica}}(\mathbf{U})[\xi]$ is given by

$$\operatorname{Degrad} f_{\text{ica}}(\mathbf{U})[\xi] = -\frac{1}{n}\sum_{i=1}^{n} 4\mathbf{C}_i(\xi\operatorname{ddiag}(\mathbf{U}^{\top}\mathbf{C}_i\mathbf{U}) + \mathbf{U}\operatorname{ddiag}(\xi^{\top}\mathbf{C}_i\mathbf{U}) + \mathbf{U}\operatorname{ddiag}(\mathbf{U}^{\top}\mathbf{C}_i\xi)).$$

**PCA problem**: Given an orthonormal matrix projector $\mathbf{U} \in \mathrm{St}(r,d)$, which is the Stiefel manifold that is the set of matrices of size $d \times r$ with orthonormal columns, the principal components analysis (PCA) problem is to minimize the sum of squared residual errors between projected data points and the original data as

$$\min_{\mathbf{U}\in\mathrm{St}(r,d)} \frac{1}{n}\sum_{i=1}^{n}\|\boldsymbol{z}_i - \mathbf{U}\mathbf{U}^{\top}\boldsymbol{z}_i\|_2^2,$$

where $\boldsymbol{z}_i$ is a data vector of size $d \times 1$. This problem is equivalent to

$$\min_{\mathbf{U}\in\mathrm{St}(r,d)} f_{\text{pca}}(\mathbf{U}) := -\frac{1}{n}\sum_{i=1}^{n}\boldsymbol{z}_i^{\top}\mathbf{U}\mathbf{U}^{\top}\boldsymbol{z}_i.$$

Here, the critical points in the space $\mathrm{St}(r,d)$ are not isolated because the cost function remains unchanged under the group action $\mathbf{U} \mapsto \mathbf{U}\mathbf{O}$ for all orthogonal matrices $\mathbf{O}$ of size $r \times r$. Subsequently, the PCA problem is an optimization problem on the Grassmann manifold $\mathrm{Gr}(r,d)$.

Similar to the arguments in the ICA problem above, the expressions of the Riemannian gradient and Hessian for the PCA problem on the Grassmann manifold are as follows:

$$\begin{aligned}
\operatorname{grad} f_{\text{pca}}(\mathbf{U}) &= \mathrm{P}_{\mathbf{U}} \operatorname{egrad} f_{\text{pca}}(\mathbf{U}) = \mathrm{P}_{\mathbf{U}}\left(-\frac{1}{n}\sum_{i=1}^{n} 2\boldsymbol{z}_i\boldsymbol{z}_i^{\top}\mathbf{U}\right) \\
\operatorname{Hess} f_{\text{pca}}(\mathbf{U})[\xi] &= \mathrm{P}_{\mathbf{U}}\left(-\frac{2}{n}\sum_{i=1}^{n}\boldsymbol{z}_i\boldsymbol{z}_i^{\top}\xi - (\xi\mathbf{U}^{\top}+\mathbf{U}\xi^{\top})\boldsymbol{z}_i\boldsymbol{z}_i^{\top}\mathbf{U} - \mathbf{U}\mathbf{U}^T\boldsymbol{z}_i\boldsymbol{z}_i^{\top}\xi\right),
\end{aligned}$$

where the orthogonal projector $\mathrm{P}_{\mathbf{U}}(\mathbf{W}) = \mathbf{W} - \mathbf{U}\mathbf{U}^{\top}\mathbf{W}$.

**MC problem**: The matrix completion (MC) problem amounts to completing an incomplete matrix $\mathbf{Z}$, say of size $d \times n$, from a small number of entries by assuming a low-rank model for the matrix. If $\Omega$ is the set of the indices for which we know the entries in $\mathbf{Z}$, the rank-$r$ MC problem amounts to solving the problem

$$\min_{\mathbf{U}\in\mathbb{R}^{d\times r},\mathbf{A}\in\mathbb{R}^{r\times n}} \|\mathcal{P}_{\Omega}(\mathbf{U}\mathbf{A}) - \mathcal{P}_{\Omega}(\mathbf{Z})\|_F^2,$$

where the operator $\mathcal{P}_{\Omega}(\mathbf{Z}_{pq}) = \mathbf{Z}_{pq}$ if $(p,q) \in \Omega$ and $\mathcal{P}_{\Omega}(\mathbf{Z}_{pq}) = 0$ otherwise is called the orthogonal sampling operator and is a mathematically convenient way to represent the subset of known entries. Partitioning $\mathbf{Z} = [\boldsymbol{z}_1, \boldsymbol{z}_2, \ldots, \boldsymbol{z}_i]$, the problem is equivalent to the problem

$$\min_{\mathbf{U}\in\mathbb{R}^{d\times r},\boldsymbol{a}_i\in\mathbb{R}^r} \frac{1}{n}\sum_{i=1}^{n}\|\mathcal{P}_{\Omega_i}(\mathbf{U}\boldsymbol{a}_i) - \mathcal{P}_{\Omega_i}(\boldsymbol{z}_i)\|_2^2,$$

where $z_i \in \mathbb{R}^d$ and the operator $\mathcal{P}_{\Omega_i}$ is the sampling operator for the $i$-th column. Given $\mathbf{U}$, $a_i$ admits the closed-form solution $a_i = \mathbf{U}_{\Omega_1}^{\dagger} z_{i\Omega_i}$, where $\dagger$ is the pseudo inverse and $\mathbf{U}_{\Omega_i}$ and $z_{i\Omega_i}$ are respectively the rows of $\mathbf{U}$ and $z_i$ corresponding to the row indices in $\Omega_i$. Consequently, the problem only depends on the column space of $\mathbf{U}$ and is on the Grassmann manifold [9], i.e.,

$$\min_{\mathbf{U} \in \mathrm{St}(r,d)} f_{\mathrm{mc}}(\mathbf{U}) := \min_{a_i \in \mathbb{R}^r} \frac{1}{n} \sum_{i=1}^{n} \|\mathcal{P}_{\Omega_i}(\mathbf{U}a_i) - \mathcal{P}_{\Omega_i}(z_i)\|_2^2.$$

The expressions of the Riemannian gradient and Hessian for the MC problem on the Grassmann manifold are as follows:

$$\mathrm{grad} f_{\mathrm{mc}}(\mathbf{U}) = \mathrm{P_U}\, \mathrm{egrad} f_{\mathrm{mc}}(\mathbf{U}) = \mathrm{P_U}\left(\frac{1}{n}\sum_{i=1}^{n} 2(\mathcal{P}_{\Omega_i}(\mathbf{U}a_i) - \mathcal{P}_{\Omega_i}(z_i))a_i^{\top}\right)$$

$$\mathrm{Hess} f_{\mathrm{mc}}(\mathbf{U})[\xi] = \mathrm{P_U}\left(\frac{2}{n}\sum_{i=1}^{n}(\mathcal{P}_{\Omega_i}(\mathbf{U}a_i) - \mathcal{P}_{\Omega_i}(z_i))b_i^{\top} + (\mathcal{P}_{\Omega_i}(\xi a_i + \mathbf{U}b_i))a_i^{\top}\right),$$

where the orthogonal projector $\mathrm{P_U}(\mathbf{W}) = \mathbf{W} - \mathbf{U}\mathbf{U}^{\top}\mathbf{W}$. Here $a_i = \mathbf{U}_{\Omega_1}^{\dagger} z_{i\Omega_i}$ and $b_i$ is the directional derivative of $a_i$ along $\xi$ and is the solution to the linear equation

$$\mathbf{U}_{\Omega_i}^{\top}\mathbf{U}_{\Omega_i}b_i = \xi_{\Omega_i}^{\top}z_{i\Omega_i} - (\xi_{\Omega_i}^{\top}\mathbf{U} + \mathbf{U}^{\top}\xi_{\Omega_i})a_i.$$

# B Proofs of Theorems

## B.1 Proof of Theorem 3.1

**Lemma B.1.** *Under Assumptions 1, 2, and 3, we have*

$$|\hat{m}_k(\eta_k) - \hat{f}_k(\eta_k)| \leq \frac{1}{2}L_H\Delta_t^3 + \delta_g\Delta_t + \frac{1}{2}\delta_H\Delta_t^2.$$

*Proof.* The absolute difference between $\hat{m}_k(\eta_k)$ and $\hat{f}_k(\eta_k)$ is bounded as below;

$$|\hat{m}_k(\eta_k) - \hat{f}_k(\eta_k)|$$

$$= \left| f(x_k) + \langle G_k, \eta_k \rangle_{x_k} + \frac{1}{2}\langle \eta_k, H_k[\eta_k] \rangle_{x_k} - \hat{f}_k(\eta_k) \right|$$

$$= \left| \hat{f}_k(\eta_k) - f(x_k) - \langle G_k, \eta_k \rangle_{x_k} - \frac{1}{2}\langle \eta_k, H_k[\eta_k] \rangle_{x_k} \right|$$

$$= \left| \hat{f}_k(\eta_k) - f(x_k) - \langle \mathrm{grad} f(x_k), \eta_k \rangle_{x_k} - \frac{1}{2}\langle \eta_k, \nabla^2\hat{f}_k(0_{x_k})[\eta_k] \rangle_{x_k} \right.$$

$$\left. + \langle \mathrm{grad} f(x_k), \eta_k \rangle_{x_k} - \langle G_k, \eta_k \rangle_{x_k} + \frac{1}{2}\langle \eta_k, \nabla^2\hat{f}_k(0_{x_k})[\eta_k] \rangle_{x_k} - \frac{1}{2}\langle \eta_k, H_k[\eta_k] \rangle_{x_k} \right|$$

$$\leq \left| \hat{f}_k(\eta_k) - f(x_k) - \langle \mathrm{grad} f(x_k), \eta_k \rangle_{x_k} - \frac{1}{2}\langle \eta_k, \nabla^2\hat{f}_k(0_{x_k})[\eta_k] \rangle_{x_k} \right|$$

$$+ |\langle \mathrm{grad} f(x_k) - G_k, \eta_k \rangle_{x_k}| + \left| \frac{1}{2}\langle \eta_k, \nabla^2\hat{f}_k(0_{x_k})[\eta_k] \rangle_{x_k} - \frac{1}{2}\langle \eta_k, H_k[\eta_k] \rangle_{x_k} \right|$$

$$\leq \frac{1}{2}L_H\|\eta_k\|_{x_k}^3 + \delta_g\|\eta_k\|_{x_k} + \frac{1}{2}\delta_H\|\eta_k\|_{x_k}^2$$

$$\leq \frac{1}{2}L_H\Delta_t^3 + \delta_g\Delta_t + \frac{1}{2}\delta_H\Delta_t^2,$$

where the first inequality uses the Cauchy-Schwarz inequality and the second one uses Assumptions 2 and 4. This completes the proof. $\qquad\square$

The proof of Theorem 3.1 follows that of [36, 37]. Therefore, this section gives its sketch.

*Proof.* Given Assumptions 1, 2, 3, 4, 5, and 6, and suppose $\|G_k\|_{x_k} \geq \epsilon_g$ and the bounds of

$$\Delta_k \leq \min \left\{ \frac{\epsilon_g}{1 + K_H}, \sqrt{\frac{(1 - \rho_{TH})\epsilon_g}{12 L_H}}, \frac{(1 - \rho_{TH})\epsilon_g}{3} \right\},$$

then we first show that the iteration $k$ is successful, i.e., $\Delta_{k+1} = \gamma \Delta_k$. For this proof, the bound of $|\hat{m}_k(\eta_k) - \hat{f}_k(\eta_k)|$ in Lemma B.1 is used.

On the other hand, for the case $\|G_k\|_{x_k} < \epsilon_g$ and $\lambda_{\min}(H_k) < -\epsilon_H$, we have $\hat{m}_k(\eta) = f(x_k) + \frac{1}{2}\langle \eta_k, H_k[\eta_k]\rangle_{x_k}$ from (2), and $\hat{m}_k(0_{x_k}) - \hat{m}_k(\eta_k) \geq \hat{m}_k(0_{x_k}) - \hat{m}_k(\eta_k^E) \geq \frac{1}{2}\nu|\lambda_{min}(H_k)|\Delta_k^2$ from Assumption 5. Then, if we have

$$\delta_H < \frac{1 - \rho_{TH}}{2}\nu\epsilon_H \quad \text{and} \quad \Delta_k \leq (1 - \rho_{TH})\frac{\nu\epsilon_H}{L_H},$$

the iteration $k$ is successful, i.e., $\Delta_{k+1} = \gamma \Delta_k$.

Combining the two above, we have for all $k$

$$\Delta_k \geq \frac{1}{\gamma}\min\left\{ \frac{\epsilon_g}{1 + K_H}, \sqrt{\frac{(1 - \rho_{TH})\epsilon_g}{12 L_H}}, \frac{(1 - \rho_{TH})\epsilon_g}{3}, \frac{\nu\epsilon_H}{L_H} \right\}$$

under Assumption 6. Consequently, we obtain the upper bound of successful iterations $|N_{\text{succ}}|$ is as $|N_{\text{succ}}| \leq \frac{f(x_0) - f(x^*)}{C\epsilon_H \min\{\epsilon_g^2, \epsilon_H^2\}}$, where $C$ is a constant depending on $L_H, K_H, \delta_g, \delta_H, \rho_{TH}$, and $\nu$. Subsequently, we obtain the claim. $\qquad\square$

## B.2  Proof of Theorem 4.1

This section gives the proof of Theorem 4.1. For this purpose, we introduce the vector Bernstein inequality for completeness before the actual proof. It should be noted that, since the retraction is a second-order retraction, we have the Hessian agreement, i.e., $\mathrm{Hess}f(x) = \nabla^2 \hat{f}_k(0_{x_k})$. In addition, it should be also noted that we assume for simplicity (and without loss of any generality) that all representations of points on the manifold, e.g., the Riemannian gradient, are vectors throughout the analysis.

**Lemma B.2** (Vector Bernstein inequality [54, 55, 38]). *Let $A_1, \ldots, A_n$ be independent random vector-valued variables with common dimension $d$ and assume that each one is centered, uniformly bounded and also that the variance is bounded above as $\mathbb{E}[A_i] = 0$, $\|A_i\|_2 \leq \mu$ and $\|\mathbb{E}[A_i^2]\|_2 \leq \sigma^2$ for positive constants $\mu$ and $\sigma$. In addition, let $Z$ be the sum of $A_i$ as $Z = \frac{1}{n}\sum_{i=1}^n A_i$. Then, we have for $0 < \epsilon < \sigma^2/\mu$*

$$\Pr(\|Z\|_2 \geq \epsilon) \leq \exp\left( -n \cdot \frac{\epsilon^2}{8\sigma^2} + \frac{1}{4} \right).$$

Now, we give the proof of Theorem 4.1.

*Proof.* The first part is for the bound of $|\mathcal{S}_g|$. We consider $|\mathcal{S}_g|$ random matrices $G_j(x)$ for $j = 1, 2, \ldots, |\mathcal{S}_g|$, where we have

$$\Pr(G_j(x) = \mathrm{grad}f_j(x)) = \frac{1}{n}.$$

We define $X_j$ as

$$X_j \triangleq G_j(x) - \mathrm{grad}f(x), \qquad j = 1, 2, \ldots, |\mathcal{S}_g|.$$

It should be noted that, since $G_j(x)$ is a randomly selected matrix, the expectation of the matrix $X_j$ should be zero, i.e., $\mathbb{E}[X_j] = 0$. Then, we define $X$ as

$$X \triangleq \frac{1}{|\mathcal{S}_g|}\sum_{j=1}^{|\mathcal{S}_g|} X_j = \frac{1}{|\mathcal{S}_g|}\sum_{j=1}^{|\mathcal{S}_g|}(G_j(x) - \mathrm{grad}f(x))$$

Selecting as $G_j(x) = \operatorname{grad} f_1(x)$ and addressing $\mathbb{E}[X_j] = 0$, we have

$$
\begin{aligned}
\|X_j^2\|_2 \ \leq \ \|X_j\|_2^2 \ &= \ \|\operatorname{grad} f_1(x) - \operatorname{grad} f(x)\|_2^2 \\
&= \ \|\operatorname{grad} f_1(x) - \frac{1}{n} \sum_{i=1}^{n} \operatorname{grad} f_i(x)\|_2^2 \\
&= \ \left\| \frac{n-1}{n} \operatorname{grad} f_1(x) - \frac{1}{n} \sum_{i=2}^{n} \operatorname{grad}_i(x) \right\|_2^2 \\
&\leq \ 2 \left( \frac{n-1}{n} \right)^2 \|\operatorname{grad} f_1(x)\|_2^2 + 2 \left( \frac{1}{n} \right)^2 \left\| \sum_{i=2}^{n} \operatorname{grad}_i(x) \right\|_2^2 \\
&\leq \ 2 \left( \frac{n-1}{n} \right)^2 (K_g^{\max})^2 + 2 \left( \frac{1}{n} \right)^2 \|(n-1) K_g^{\max}\|_2^2 \\
&= \ 4 \left( \frac{n-1}{n} \right)^2 (K_g^{\max})^2 \ \leq \ 4 (K_g^{\max})^2,
\end{aligned}
$$

where the first inequality uses $(a+b)^2 \leq 2a^2 + 2b^2$.

Now, we apply the vector Bernstein inequality in Lemma B.2 replacing $Z$ with $X$, we obtain

$$
\begin{aligned}
\Pr \left( \left\| \frac{1}{|\mathcal{S}_g|} \sum_{j=1}^{|\mathcal{S}_g|} G_j(x) - \operatorname{grad} f(x) \right\|_2 \geq \epsilon \right) \ &= \ \Pr \left( \|X\|_2 \geq \epsilon \right) \\
&\leq \ \exp \left( \frac{-\epsilon^2 |\mathcal{S}_g|}{32 (K_g^{\max})^2} + \frac{1}{4} \right).
\end{aligned}
$$

Here, we require the probability that the approximate deviation of the sub-sampled gradient from the exact $\operatorname{grad} f(x)$ is higher than $\epsilon$ to be lower than some $\delta \in (0, 1]$, we have

$$
\exp \left( \frac{-\epsilon^2 |\mathcal{S}_g|}{32 (K_g^{\max})^2} + \frac{1}{4} \right) \ = \ \delta \qquad \Longrightarrow \qquad \epsilon \ = \ 4\sqrt{2} K_g^{\max} \sqrt{\frac{\log(1/\delta) + 1/4}{|\mathcal{S}_g|}}.
$$

From Assumption 4, we finally obtain

$$
\begin{aligned}
\|G_k - \operatorname{grad} f(x_k)\|_2 \ &\leq \ \delta_g \\
\Longrightarrow 4\sqrt{2} K_g^{\max} \sqrt{\frac{\log(1/\delta) + 1/4}{|\mathcal{S}_g|}} \ &\leq \ \delta_g \\
\Longrightarrow \qquad |\mathcal{S}_g| \ &\geq \ \frac{32 (K_g^{\max})^2 (\log(1/\delta) + 1/4)}{\delta_g^2}.
\end{aligned}
$$

Next, we consider $|\mathcal{S}_H|$ random matrices $H_j(x)$ for $j = 1, 2, \ldots, |\mathcal{S}_H|$. For this purpose, we denote the $j$-th element of $\nabla^2 \hat{f}(0_x)$ for the $j$-th sample as $\nabla^2 \hat{f}_j(0_x)$. Similarly to the case above, we assume the uniform sampling strategy as $\Pr(H_j(x) = \nabla^2 \hat{f}_j(0_x)) = \frac{1}{n}$. Now, for $\eta \in T_x \mathcal{M}$, we define $Y_j$ as

$$
Y_j \ \triangleq \ H_j(x)[\eta] - \nabla^2 \hat{f}(0_x)[\eta], \qquad j = 1, 2, \ldots, |\mathcal{S}_H|.
$$

It should be noted that, since $H_j(x)$ is randomly selected and $\eta$ is independent of $H_j(x)$, the expectation of the matrix $Y_j$ should be zero, i.e., $\mathbb{E}[Y_j] = 0$. Then, we define $Y$ as

$$
Y \ \triangleq \ \frac{1}{|\mathcal{S}_H|} \sum_{j=1}^{|\mathcal{S}_H|} Y_j \ = \ \frac{1}{|\mathcal{S}_H|} \sum_{j=1}^{|\mathcal{S}_H|} \left( H_j(x)[\eta] - \nabla^2 \hat{f}(0_x)[\eta] \right)
$$

Then, for $\nabla^2 \hat{f}_1(0_x)$, we have

$$
\begin{aligned}
\|Y_j^2\|_2 \ \leq \ \|Y_j\|_2^2 \ &= \ \left\| \frac{n-1}{n} \nabla^2 \hat{f}_1(0_x)[\eta] - \frac{1}{n} \sum_{i=2}^{n} \nabla^2 \hat{f}_i(0_x)[\eta] \right\|_2^2 \\
&\leq \ 4 \left( \frac{n-1}{n} \right)^2 (K_H^{\max})^2 \|\eta\|_2^2 \ \leq \ 4(K_H^{\max})^2 \|\eta\|_2^2.
\end{aligned}
$$

Now, we apply the vector Bernstein inequality in Lemma B.2. Similarly to the sub-sampled gradient, we obtain

$$
\Pr \left( \left\| \frac{1}{|\mathcal{S}_H|} \sum_{j=1}^{|\mathcal{S}_H|} H_j(x)[\eta] - \nabla^2 \hat{f}(0_x)[\eta] \right\|_2 \geq \epsilon \right) \ \leq \ \exp \left( \frac{-\epsilon^2 |\mathcal{S}_H|}{32(K_H^{\max})^2 \|\eta\|_2^2} + \frac{1}{4} \right).
$$

Then, we obtain $\epsilon = 4\sqrt{2} K_H^{\max} \|\eta\|_2 \sqrt{\frac{\log(1/\delta)+1/4}{|\mathcal{S}_H|}}$. From Assumption 4, we finally obtain

$$
\begin{aligned}
\|(H_k - \nabla^2 \hat{f}(0_x))[\eta]\|_2 \ &\leq \ \delta_H \|\eta\|_2 \\
\implies 4\sqrt{2} K_H^{\max} \|\eta\|_2 \sqrt{\frac{\log(1/\delta)+1/4}{|\mathcal{S}_H|}} \ &\leq \ \delta_H \|\eta\|_2 \\
\implies \qquad |\mathcal{S}_H| \ &\geq \ \frac{32(K_H^{\max})^2 \log(1/\delta)+1/4}{\delta_H^2}.
\end{aligned}
$$

This completes the proof. $\qquad\qquad\qquad\qquad\qquad\qquad\qquad\qquad\qquad\qquad\qquad\qquad\qquad\qquad$ $\square$

# C   Additional numerical comparisons

In this section, we show additional numerical comparisons which do not appear in the main paper.

## C.1   PCA problem

Additional results of different runs for **Cases P1**, **P2**, **P3**, and **P4** are shown in Figure A.1.

(i-1) Run 2: Oracle calls.    (i-2) Run 2: Run time.    (ii-1) Run 3: Oracle calls.    (ii-2) Run 3: Run time.

(a) **Case P1**

(i-1) Run 2: Oracle calls.    (i-2) Run 2: Run time.    (ii-1) Run 3: Oracle calls.    (ii-2) Run 3: Run time.

(b) **Case P2**

(i-1) Run 2: Oracle calls.    (i-2) Run 2: Run time.    (ii-1) Run 3: Oracle calls.    (ii-2) Run 3: Run time.

(c) **Case P3**

(i-1) Run 2: Oracle calls.    (i-2) Run 2: Run time.    (ii-1) Run 3: Oracle calls.    (ii-2) Run 3: Run time.

(d) **Case P4**

Figure A.1: Performance evaluations on PCA problem (**Case P1**, **P2**, **P3**, **P4** ).

Additional results of different runs for **Case P6** are shown in Figure A.2.

(i-1) Run 2: Oracle calls.    (i-2) Run 2: Run time.    (ii-1) Run 3: Oracle calls.    (ii-2) Run 3: Run time.

(a) **Case P6**

Figure A.2: Performance evaluations on the PCA problem (**Case P6**).

## C.2    MC problem

Additional results of different runs for **Cases M1**, **M2**, **M3**, **M4**, and **M5** are shown in Figures A.3, A.4, A.5, A.6, and A.7, respectively.

(i) Oracle calls vs. Train MSE    (ii) Run time vs. Train MSE    (iii) Oracle calls vs. Test MSE    (iv) Run time vs. Test MSE

(a) Run 2

(i) Oracle calls vs. Train MSE    (ii) Run time vs. Train MSE    (iii) Oracle calls vs. Test MSE    (iv) Run time vs. Test MSE

(b) Run 3

Figure A.3: Performance evaluations on the MC problem (**Case M1**).

(i) Oracle calls vs. Train MSE    (ii) Run time vs. Train MSE    (iii) Oracle calls vs. Test MSE    (iv) Run time vs. Test MSE

(a) Run 2

(i) Oracle calls vs. Train MSE    (ii) Run time vs. Train MSE    (iii) Oracle calls vs. Test MSE    (iv) Run time vs. Test MSE

(b) Run 3

Figure A.4: Performance evaluations on the MC problem (**Case M2**).

(i) Oracle calls vs. Train MSE    (ii) Run time vs. Train MSE    (iii) Oracle calls vs. Test MSE    (iv) Run time vs. Test MSE

(a) Run 2

(i) Oracle calls vs. Train MSE    (ii) Run time vs. Train MSE    (iii) Oracle calls vs. Test MSE    (iv) Run time vs. Test MSE

(b) Run 3

Figure A.5: Performance evaluations on the MC problem (**Case M3**).

(i) Oracle calls vs. Train MSE    (ii) Run time vs. Train MSE    (iii) Oracle calls vs. Test MSE    (iv) Run time vs. Test MSE

(a) Run 2

(i) Oracle calls vs. Train MSE    (ii) Run time vs. Train MSE    (iii) Oracle calls vs. Test MSE    (iv) Run time vs. Test MSE

(b) Run 3

Figure A.6: Performance evaluations on the MC problem (**Case M4**).

(i) Oracle calls vs. Train MSE    (ii) Run time vs. Train MSE    (iii) Oracle calls vs. Test MSE    (iv) Run time vs. Test MSE

(a) Run 2

(i) Oracle calls vs. Train MSE    (ii) Run time vs. Train MSE    (iii) Oracle calls vs. Test MSE    (iv) Run time vs. Test MSE

(b) Run 3

Figure A.7: Performance evaluations on the MC problem (**Case M5**).