[Reviews · NeurIPS 2018]

Reviewer 1



Summary: This paper proposes an inexact trust-region method for Riemannian optimization, for the purpose of finding an approximate second-order stationary point for large scale finite-sum optimization problems. The proposed algorithm and analysis are based on existing work in vector space inexact trust-region as well as Riemannian trust-region methods. The authors also perform experiments in PCA and matrix completion as an empirically support for the proposed algorithm. Pros: This work seems to be a solid addition to the Riemannian optimization repertoire. Both the analysis and the experiments show merits of the algorithm. The analysis is new but considering previous work, it is somewhat incremental. Cons: 1) Some of the details of the experiments are missing. For example, in the PCA experiments, RSVRG seems to be competitive in terms of oracle calls but fall behind in terms of wall-clock time. Is there a particular reason for this difference? e.g. are efficient retractions/vector transports used? 2) In the analysis you made assumptions about the pullback functions (e.g. Lipschitz Hessian). It would be good to discuss how such assumptions hold in practice (e.g. bound on the Lipschitz constant, etc.) for problems of interest.

Reviewer 2



This paper proposed the inexact TR optimization algorithm on manifold, in which the analysis largely follows from the euclidean case [27,28]. They also considered the subsampling variant of their method for finite-sum problems and the empirical performance on PCA and matrix completion problems. 1. It seems to me the convergence analysis follows exactly from the works [27,28] except for some different notation. At least this paper doesn't show what is the technical challenge for the Riemannian case. 2. For the experiments, both MC and PCA problems can be solved efficiently by Euclidean method like GD/SGD. What are the advantages of using Riemannian optimization method? 3. In appendix, line 410, C should not depend on \delta_g or \delta_H otherwise it will depend on \eps_g and \eps_H. Overall, it is a good paper with very interesting results. The presentation can be further improved. First, show some real motivation for doing inexact RTR method. Right now the intro is a little bit general and vague. Second, show the technical challenges/difficulties of obtaining the iteration complexity results and the analysis should be elaborated a little bit. Third, show some background/examples on how to compute the gradient and Hessian on manifold, i.e. how you can do it efficiently in practice for certain problems.

Reviewer 3



This paper proposes an inexact variant fo Riemannian trust region algorithm, which is usually computationally expensive due to the Hessian term. However, I’m not an expert in this specific area and I do have some difficulties appreciating the originality and importance of this paper. It’s true that calculating the full hessian in a Riemannian trust region algorithm is expensive, but there are a lot of efficient alternatives already. (I was referring to other numerical algorithms that escapes saddles rather than RTR.) Technically, I found the first assumption made in this paper might be hard to verify in practice. Could the author give some justification for the applicability of this assumption?(Thanks for the clarification) Theorem 4.1 claims that at any $x_k$, assumption 2 are satisfied with high probability. From my understanding, for the sub-sampled algorithm to perform as desired, we want the assumption to be satisfied at any iteration $k$. There seems to be a gap here. Could the author give me some clarification on this? (I went through the proof more carefully, and I was confused the first time.) The numerical experiment part is sufficient and serves the goal.